# The Double Edged Sword: Identifying Authentication Pages and their Fingerprinting Behavior

## ABSTRACT

Browser fingerprinting is often associated with cross-site user tracking, a practice that many browsers (e.g., Safari, Brave, Edge, Firefox, and Chrome) want to block. However, less is publicly known about its uses to enhance online safety, where it can provide an additional security layer against service abuses (e.g., in combination with CAPTCHAs) or during user authentication. To the best of our knowledge, no fingerprinting defenses deployed thus far consider this important distinction when blocking fingerprinting attempts, so they might negatively affect website functionality and security.

To address this issue we make three main contributions. First, we propose and evaluate a novel machine learning-based method to automatically identify authentication pages (i.e. login and sign-up pages). Our supervised algorithm achieves 96-98% precision and recall on a large, manually-labelled dataset of 10,000 popular sites. Second, we compare our algorithm with other methods from prior works on the same dataset, showing that it significantly outperforms all of them (+83% F1-score). Third, we quantify the prevalence of fingerprinting scripts across login and sign-up pages (9.2%) versus those executed on other pages (8.9%); while the rates of fingerprinting are similar, home pages and authentication pages differ in the third-party scripts they include and how often these scripts are labeled as tracking. We also highlight the substantial differences in fingerprinting behavior on login and sign-up pages. Our work sheds light on the complicated reality that fingerprinting is used to both protect user security and invade user privacy, and that this dual nature must be considered by fingerprinting mitigations.

## 1 INTRODUCTION

In today's digital landscape, browser fingerprinting has garnered attention for its use tracking individuals' online activities. Doing so involves collecting a set of attributes from a user's web browser and device in order to derive a unique identifier that persists across different websites. Some of the attributes that could be used for creating a browser fingerprint include browser features and configurations (e.g. the User-Agent string, the canvas API, installed plugins), OS features (e.g. emoji sets), and even hardware features (e.g. battery level [28]). A 2021 study found that approximately 10% of sites perform fingerprinting [33]. This rate may increase over time, since the deprecation of third-party cookies [42, 50, 54] might nudge online trackers to switch to cookieless alternatives, such as fingerprinting.

What makes fingerprinting attractive for tracking – the ability to uniquely identify a device – also gives it potential for enhancing security. Consider the case of an unauthorized user logging into a victim's account with the correct username and password, but a different fingerprint. The site could send a multi-factor authentication (MFA) prompt to the victim. This approach not only enhances security but also minimizes user inconvenience by forgoing additional

hardware and complex authentication procedures. Beyond preventing account compromise, fingerprinting can also help web services detect bots and thus prevent click fraud and cookie hijacking [27]. In the absence of privacy-friendly alternatives, it is important to take into account the context in which fingerprinting occurs when making an enforcement decision against it.

Durey et al. studied the usage of browser fingerprinting as a means to enhance web security [27]. They manually analyzed four page categories – login, sign-up, payment, and shopping cart – across 1,485 pages from 446 domains. While their work provided an important initial set of results, it suffered from two main limitations. First, it relied solely on a manual analysis of websites, which is not scalable nor generalizable. Second, it did not consider the impact of any existing anti-fingerprinting tool on such websites.

To address these limitations, we developed a novel machine learning (ML) model to identify login and sign-up pages, and we present a large-scale empirical study on the usage of browser fingerprinting on these pages. Our model, which achieves a greater than 96% precision and recall for each page category, can run on-device for on-the-fly inference, as demonstrated by our Chrome extension that we will publicly release. Many studies analyze login and sign-up pages [25, 46, 53], predominantly relying on heuristic methods for detection in a centralized, server-based setting. However, given the dynamic nature of the web, rule-based detection techniques quickly become outdated (e.g. login and sign-up pages with multi-step designs or lacking password fields) and thus more dynamic approaches, such as our ML-based solution, are necessary.

Our study makes the following key contributions:

- A web measurement study evaluating browser fingerprinting for security on the top 100,000 websites' login and sign-up pages. Using an instrumented crawler, we detect fingerprinting attempts and potential login/sign-up indicators.
- An ML model identifying login and sign-up pages with high precision and recall (96-98%).
- A browser extension for identifying and displaying login and sign-up pages, along with a web crawler for listing such pages in given URLs, both of which will be publicly available.

Our results show that 9.2% of login and sign-up pages perform fingerprinting, compared to 8.9% across all pages. While the rates are similar, fingerprinting scripts on home pages are more likely to be classified as trackers and far more likely to perform canvas fingerprinting than scripts on login and sign-up pages. For sites that fingerprint on at least one authentication page, 50% of them fingerprint only on the login page. When sites fingerprint on both login and sign-up pages, they use scripts from the same set of third-parties in 98% of cases. These new findings show the multifaceted intent behind fingerprinting scripts on the web.

## 2 BACKGROUND AND RELATED WORK

In this section we provide background on fingerprinting, including its use for authentication, and describe prior work on identifying login and sign-up pages.

### 2.1 Browser Fingerprinting & Mitigations

Browser fingerprinting is a method that sites use to generate a unique identifier that can link the same browser across different domains or visits. It is derived by joining multiple pieces of information about the user's browser, typically via HTTP headers and JavaScript APIs. Fingerprinting is effective for tracking because it is stateless, less visible to the users, and – unlike tracking via cookies – difficult to disable. For example, the library may learn about the user's timezone and the list of fonts they have installed on their system. As the library collects more information, it can potentially uniquely distinguish a user among millions of visitors to a given website.

Mitigations for fingerprinting broadly rely on four approaches: randomization, normalization, heuristics, and machine learning. Randomization methods, like Privaricator [43] and FPRandom [36], add noise to APIs like canvas so that the same user presents different fingerprints during different sessions. However, adding noise may affect API functionality and is reversible, potentially serving as a fingerprint itself. Another approach, normalization, aims to standardize fingerprints for multiple users and is implemented by the Tor and Brave browsers [1, 10].

While randomization and normalization attempt to disrupt fingerprinting scripts, the next two approaches try to identify fingerprinting to block it entirely. Heuristic-based identification methods like Privacy Badger [8], JShelter [7], and Disconnect [3] rely on predefined rules, which can miss some fingerprinting scripts and require continuous updates. Learning-based methods such as FP-Inspector [33] are more effective but can suffer from a higher false positive rate than heuristic-based methods.

### 2.2 Fingerprinting for Authentication

Many studies have proposed using browser fingerprinting as an additional authentication technique [20, 21, 32, 45, 47, 51]. Doing so can protect users whose credentials are stolen; if an attacker gets a user's credentials, the website can detect that the attacker has a different fingerprint than the victim. Then the site could show a multi-factor authentication (MFA) prompt to the attacker to protect the user. Prior work has also proposed using fingerprinting to protect against cookie hijacking by detecting when cookies are used on a device with a different fingerprint [27], and to quickly identify bots [27, 55].

Our work builds on the analysis from Durey et al. [27], which manually analyzes 1,485 pages from 446 domains to detect browser fingerprinting on a variety of page categories, including login and sign-up pages. Login and sign-up pages each compose 12-13% of the pages they analyze, and their classifier flags 23.4% and 31.1% of login and sign-up pages as performing fingerprinting, respectively. Finally, they analyze 14 scripts developed for security and find four that are used exclusively on login and sign-up pages for payment platforms, fraud prevention, and bot detection. We build on this paper by analyzing a larger set of websites and performing automated

analysis; we also analyze the most popular websites to understand the prevalence of fingerprinting in a more general context.

We also build on the work of Lin et al. [38], which presents and evaluates an attack that uses fingerprints to bypass MFA. They detect login pages using the approach from Drakonis et al. [26], which we evaluate in Section 3.1.1. They detect login pages for 11,527 of the Alexa top 20K websites and find that the majority of these sites perform some fingerprinting. We add to these results by using a more sophisticated login page detection technique and analyzing a larger set of websites (100K vs 20K). We compare findings in more detail in §4.3. Lin et al. was one of the first to present concrete security vulnerabilities that stem from using fingerprinting for authentication; we provide a more comprehensive overview of how fingerprinting is used for authentication in the broader web ecosystem.

### 2.3 Login/Sign-Up Page Detection

Several studies have tried to identify and analyze authentication pages [19, 24, 26, 27, 31, 35, 39, 46, 52, 57]. Most of the approaches rely either on manual inspection [27, 46] or on heuristics based on regex patterns [24, 26, 31, 35, 52, 57]. Specifically, such regex patterns often include variations of the terms "login" and "sign-up" and include translations to other languages. Some of these studies used additional heuristics in addition to the regex strings, such as checking the visibility of elements and the types of input elements in forms (e.g. number of password fields) [26, 35]. Other studies queried search engines to discover authentication pages for a given domain [31, 35, 52].

However, these heuristics often fail to detect complex authentication flows, such as multi-step login flows (as illustrated in Figure 6(a)). In this example, the login form only contains a username field and requires the user to click through before showing a password field. There are similar multi-step flows for sign-up forms (see Figure 6(b)). In addition, heuristics and regex patterns can lead to misclassification. For example, some heuristics from prior work classify forms with multiple password elements as sign-up forms [26], or forms with at least three visible input fields as sign-up forms [35]. The form in Figure 6(c) is a login form that would be misclassified by these heuristics. Regex patterns can similarly misclassify newsletter forms as sign-up forms, as shown in Figure 6(d).

Instead of relying on heuristics and regex patterns, two studies trained machine learning models [19, 39]. The feature sets used by both studies tried to capture three key components: 1) the presence of login/sign-up keywords, 2) the number of password input fields, and 3) the total number of form input fields. For instance, Al Roomi and Li [19] achieved 94.5%/96.3% precision/recall for login forms, and 77.1%/99.5% precision/recall for sign-up forms, while Lodrant [39] achieved 71% accuracy. Both works found it challenging to detect multi-step authentication forms.

## 3 METHODOLOGY

In this section we describe the methodology we used to collect the website data via an instrumented crawler, how we extracted features to train the ML model to detect login and sign-up pages, and the methods we used to detect fingerprinting on such pages.

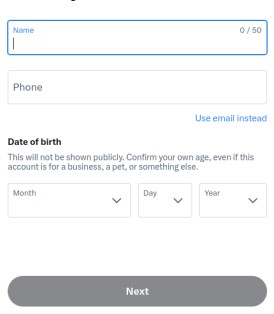
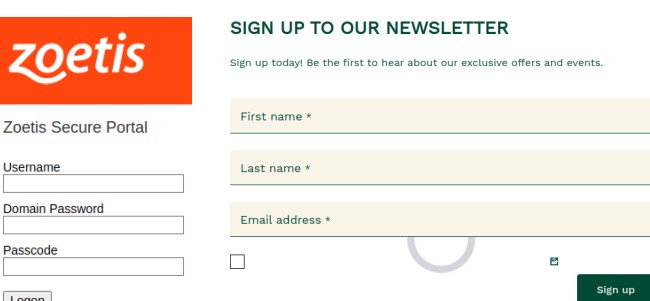

(a) Twitter's login form without password field.

(b) Twitter's sign-up form without password field.

(c) Zoetis' login form without email and with two password fields.

(d) The newsletter form on thebodyshop.com closely resembles the account registration form.

**Figure 1: Example web forms: ⓐ is a login form without a password field; ⓑ is a sign-up form without a password field; ⓒ is a login form without an email field; and ⓓ is a newsletter sign-up form that resembles an account creation form.**

## 3.1 Login/Sign-Up Page Detection

In this section, we describe the four different mechanisms used to identify login and sign-up pages.

*3.1.1 Prior Work Heuristics.* Automated discovery of login and sign-up pages is an area of study in prior works. We implement heuristics from one study by Drakonakis, Ioannidis, and Polakis into our crawler, as their code is publicly available [26].[1] Their methodology identifies login and sign-up pages using a combination of regex string searching (for English phrases such as "register," "login," and "my profile") and heuristics based on DOM elements. For example, the latter includes the number of password elements, the presence of input elements for phone numbers or dates, and the visibility of these input elements. This methodology has also been used by Lin et al. in their preliminary analysis of the use of fingerprinting on login and sign-up pages [38].

*3.1.2 Autofill Heuristics.* Autofill is a Chrome feature that automatically generates new passwords when the user visits a sign-up form. Users can also opt to save their credentials with Chrome so that Autofill can automatically fill form fields on behalf of the user. While Autofill has been studied for its security risks [18, 37, 44] and its impact on developers [41], to the best of our knowledge, it has not been studied as a tool for automated login/sign-up form detection.

```
<form>
  <input type="text" name="username" pm_parser_annotation
    ="username_element">
  <input type="password" name="password"
    pm_parser_annotation="new_password_element">
</form>
```

**Listing 1: Example of Autofill annotations for a web form**

We collect Autofill information by enabling the `show-autofill-signatures` Chrome flag, which adds HTML attributes (called `pm_parser_annotation`) to each form input element. An example is shown in Listing 1. The Autofill annotations are only available

to Puppeteer crawlers in the "new" headless mode, which includes the code in the `//chrome` path (which was previously included in headful crawlers but not "old" headless crawlers) where Autofill is implemented [23]. These annotations only include Autofill's client-side heuristics; Autofill also has a server-side component [9] but it is inaccessible outside of Google. So, the client-side heuristics may not perfectly match a user's experience with the Autofill feature.

Once the annotations are collected, classification is simple: if we see annotations for creating a new password (`new_password_element`) or confirming a password (`confirmation_password_element`), then we classify the form as a sign-up form. Otherwise, if there are annotations for a username field (`username_element`) and a password field (`password_element`), we classify the form as a login form.

*3.1.3 Fathom-based Login & Sign-up Classifier.* Fathom is a supervised-learning framework developed by Mozilla for identifying various components of web pages[29]. Their repository of rulesets showcases the integration of multiple ML models designed to detect various types of web page elements, such as pop-ups and even specific HTML components like price tags [30]. Fathom has developed classifiers for login and sign-up pages [14, 16], which we integrated into our web crawler without any modifications.

*3.1.4 Login & Sign-up Classifier.* In the following sections, we describe how we generate the test, validation, and training datasets, as well as the login and sign-up page classifier depicted in Figure 2.

**Page Type Identification & Feature Extraction.** We used the Chrome User Experience Report (CrUX) [2] to compile a list of the top 100 sites and manually browsed them to find the login and sign-up pages, if present. We manually reviewed the source code of these pages and extracted 88 different features related to distinct aspects of the design and interaction modalities. For instance, we created a regex pattern that includes many variations of terms like "login" and "sign-up" as well as their translations into several other languages; we checked form attributes, button text content and attributes, header attributes, and other HTML elements for regex matches. Finally, we checked the presence of a checkbox element with a "Remember Me" pattern for login pages. We created

---

[1]The heuristics from prior work [26] are available at https://gitlab.com/kostasdrk/xdriver3-open/-/blob/master/js/scripts.js

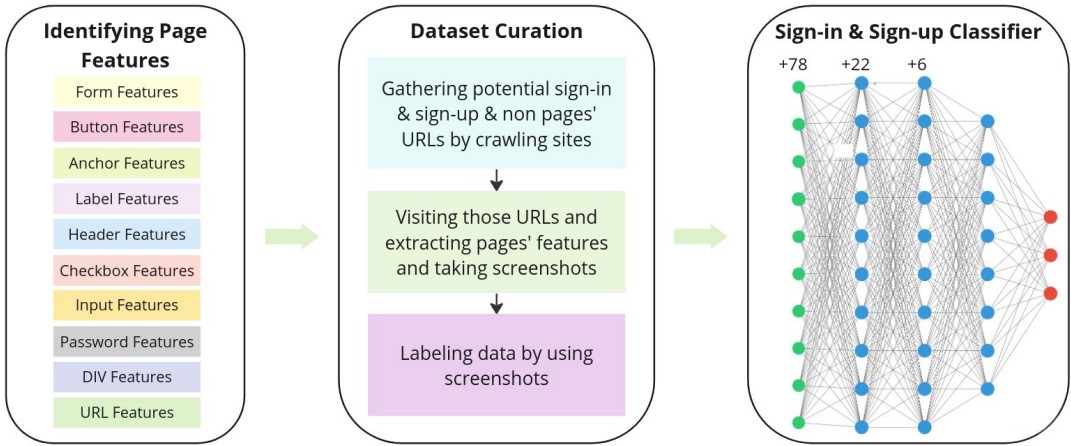

Figure 2: ML pipeline for classifying login and sign-up pages.

variations of these features that would check if they are in a form, are in an iFrame, and are visible. The full feature set is detailed in Appendix C.

**Dataset Curation.** We created a training dataset by crawling the CrUX top 10K homepages and visiting 47K other pages linked from the homepages that match the aforementioned regex string created to find login and sign-up pages (we describe this regex string and our inner page collection process in more detail in Section 3.5). We manually labeled a random sample of 1,500 login pages, 1,000 sign-up pages, and 2,500 non-authentication pages, totaling 5,000 labeled pages. However, we were unable to collect all 88 features for our model for 93 (1.9%) pages due to bot detection mechanisms; more specifically, we collected an average of 2.19 features on non-authentication pages. We filtered our dataset to only retain pages where we could collect a minimum of three features, which resulted in a dataset of 1,299 login pages, 973 sign-up pages, and 2,453 non-authentication pages for a total of 4,725 labeled pages. We split this into 67% for training and 33% for testing.

**Model Training.** We used the TensorFlow [12] framework to train a multi-class classifier. For each visited page, we generate an 88-dimensional feature vector and a label, which we fed to a neural network with two dense hidden layers containing 8 and 16 units. The output layer is mapped to the three classes, i.e., login, sign-up, and neither. We trained the model for 200 epochs using the cross-entropy loss function.

**Model Performance.** The classifier achieves high performance on the test dataset (see Table 1). Specifically, it scores a recall of 0.98 and 0.96 and a precision of 0.99 and 0.96 on login and sign-up pages, respectively. Meaning the model has an error rate of 1%-4%, and is able to correctly label the vast majority of the login and sign-up pages in the dataset.

## 3.2 Fingerprinting Detection

Browser fingerprinting has a rich history, marked by the ongoing evolution of fingerprinting and detection thereof. Prior work has used both ML-based methods [33] and heuristic approaches [28] to identify fingerprinting scripts. In this work, we relied on the

| Page Type | Accuracy | Precision | Recall | F1-score |
|-----------|----------|-----------|--------|----------|
| **Login** | 0.98 | 0.99 | 0.98 | 0.98 |
| **Sign-up** | 0.95 | 0.96 | 0.96 | 0.96 |
| **Neither** | 0.98 | 0.99 | 0.99 | 0.99 |

Table 1: Classifier performance on test dataset.

heuristics established by Englehardt and Narayanan (described in Appendix B) that monitor the Canvas, WebRTC, Canvas Font, and AudioContext APIs to detect fingerprinting scripts [28]. We logged function calls (including arguments and return values) by overriding getter and setter functions on all pages, including subframes, immediately after the document was created.

## 3.3 Crawler Implementation

Our web crawler is a fork of Tracker Radar Collector (TRC), [2] a crawler created by DuckDuckGo. TRC is built on Puppeteer and is designed to capture specific interactions with JavaScript APIs, HTTP requests and responses, cookies, and other relevant data for web measurements. We use the "new" headless mode [23] to collect Autofill signals (see Section 3.1.2).

Furthermore, we implemented a distinct collector to instrument method calls and property accesses related to fingerprinting (as explained in Section 3.2). In our approach, we modify the object's *getters* to intercept these function calls.[3]

To more closely imitate a genuine user, our crawler scrolls to the bottom of the page and back to the top, and pauses for 5 seconds before collecting data. We also experimented with filling in some form elements on the page to activate a wider range of fingerprinting scripts. We conducted two preliminary crawls on 1,000 domains: one where we filled in input fields, and one where we did not. These crawls revealed that 102 scripts attempted fingerprinting on 136

---

[2]https://github.com/duckduckgo/tracker-radar-collector
[3]Although TRC already has the capability to intercept JavaScript API calls, we introduced a separate collector due to a known TRC bug that causes it to miss the initial function calls, as detailed in a public GitHub issue [13].

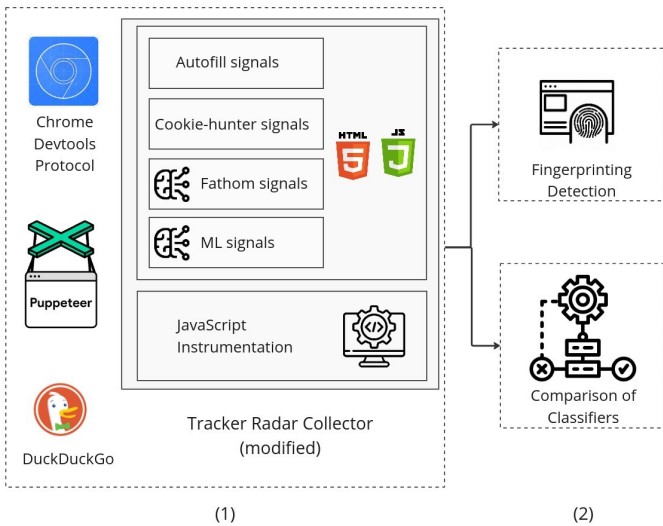

**Figure 3: (1) Our crawler extends the Tracker Radar Collector. We collect login and sign-up signals, fingerprinting signals, and JavaScript execution traces. (2) We then compare all authentication page detection methods and detect fingerprinting scripts.**

sites, regardless of whether input fields were filled or not. Since there was no difference in fingerprinting behavior, we did not fill in any form fields in subsequent crawls.

We executed both the homepage and inner page crawls using cloud-based servers provided by DigitalOcean, located in the United States. Each individual crawl was completed within a three-day timeframe, using a server with 8 vCPU cores and 16GB of RAM. The decision to use a US-based server was mainly influenced by the intention to minimize encounters with cookie consent dialogs.

### 3.4 Interaction with Consent Dialogs

After GDPR was enacted in 2018 [5], sites began to show cookie consent banners to give users transparency and control over the processing of personal information. To increase the likelihood of triggering fingerprinting scripts, we chose to consent for all forms of personal data processing, including accepting cookies. To automatically interact with cookie consent banners, we integrated code derived from Priv-Accept [34, 40], a specialized crawler designed for this purpose. Priv-Accept finds HTML elements such as <a>, <button>, and <div>, then checks for keywords such as "Accept," and triggers a click action. We ported the Priv-Accept code from Python to JavaScript without changing any logic.

### 3.5 Collection of Potential Inner Authentication Pages

Login and sign-up flows might not be displayed on the homepages but rather on dedicated inner pages. Hence, our crawler must visit both types of pages. When crawling homepages, we collected links on the homepage that may lead to inner pages with login and sign-up forms. To increase the likelihood of discovering such pages, we

employed a combined regular expression pattern used in our feature extraction (as described in Section 3.1.4). This pattern includes various word translations related to "login," "sign up," and "register." We employed this pattern search across multiple attributes of <a> elements, including `innerText`, `title`, `href`, `ariaLabel`, `placeholder`, `id`, `name`, and `className`. During this process, we filtered out URLs directing to non-HTML files, such as PDFs or images.

We validated our approach by crawling 300 of the CrUX top 1K most popular websites and checked how many login and sign-up pages our regex strings could identify. As outlined in Appendix A, our crawler initially navigated to the homepage of each website. If it found both login and sign-up pages on the homepage, it terminated; otherwise, it visited up to 15 (5 login + 5 sign-up + 5 neither) inner pages linked on the homepage that our regex pattern matched. Of the 300 websites in our sample, 4.7% had errors while loading the homepage. We successfully detected login or sign-up pages on 49% of sites, while 31% of sites did not contain such pages. In 7.7% of the remaining pages, the login and sign-up pages were only visible after interacting with an HTML element. In 4.7% cases, the login and sign-up pages were present but could not be detected by our ML model. In only 1.7% cases, the correct links were not identified using our regex pattern.

We also collected inner links that were not login or sign-up pages. We intentionally omitted links pointing to external domains. For non-login or sign-up pages, we prioritized links closer to the viewport center. This was done to prevent the collection of unrelated links located in less visible areas, such as footers. To enhance crawl efficiency, we restricted the number of inner links to five for each category.

## 4 MEASUREMENT RESULTS

Following recent best practices [48, 49], we crawled the top 100K domains from the Chrome User Experience Report (CrUX) [2] (as of April 2023) in August 2023. We only used this dataset in our subsequent analysis. We excluded 1,155 URLs on the list with identical fully qualified domain names but different schemes; as a result, we attempted to crawl 98,845 homepages and successfully visited 94,482 homepages (95.8%). After collecting crawler results (including login and sign-up signals, inner links, screenshots, and fingerprinting attempts) for the homepages, we extracted the inner links for the second round of crawling. We attempted to crawl 474,436 inner pages and successfully visited 446,688 inner pages (94.4%). Priv-Accept facilitated the acceptance of personal data processing on 26.9% of all pages crawled (including homepages and inner pages).

### 4.1 Comparison of Login/Sign-Up Detection Techniques

To compare login/sign-up detection techniques, we manually labelled a random sample of 1,000 pages (including both homepages and inner pages, based on top 100K-crawl). These 1,000 pages included 261 login pages, 160 sign-up pages, 23 pages that had both login and sign-up functionality, and 532 non-authentication pages (plus 22 pages that had errors loading). We computed precision and recall scores for each detection technique, which we show in

|  | Prior Work Heuristics | | Autofill | | Fathom | | Our ML-based solution | |
|---|---|---|---|---|---|---|---|---|
|  | Precision | Recall | Precision | Recall | Precision | Recall | Precision | Recall |
| **Login** | 0.83 | 0.51 | 0.51 | 0.75 | 0.77 | 0.79 | 0.97 | 0.89 |
| **Sign-up** | 0.58 | 0.54 | 0.47 | 0.66 | 0.36 | 0.95 | 0.83 | 0.92 |
| **Neither** | 0.71 | 0.97 | 0.80 | 0.69 | 0.88 | 0.78 | 0.92 | 0.95 |

Table 2: Assessment of login and sign-up detection methods through the analysis of a randomly sampled set of 1,000 websites.

|  | Prior Work Heuristics | Autofill | Fathom | Our ML-based solution |
|---|---|---|---|---|
| **Total login pages** | 42,375 | 91,220 | 52,307 | 52,805 |
| **Total sign-up pages** | 17,517 | 31,103 | 138,639 | 21,988 |
| **Domains with at least one login page** | 22,369 (23.68%) | 31,840 (33.70%) | 24,963 (26.42%) | 27,059 (28.64%) |
| **Domains with at least one sign-up page** | 12,199 (12.91%) | 15,620 (16.53%) | 42,672 (45.16%) | 15,998 (16.93%) |

Table 3: For each detection technique (explained in § 3), we list the number of distinct login and sign-up pages it identifies as well as the number of domains (i.e. number of CrUX list entries) it can identify a login and sign-up page for.

Table 2, and the number of pages classified as authentication pages by each technique in Table 3.

We found that our ML approach had the best precision/recall across page categories. This is, in part, due to its ability to account for multi-step designs, to validate the visibility of login and sign-up elements, and to consider their presence within iFrames. Since this approach is more effective than the others, we use it to classify login and sign-up pages for all subsequent analysis. Other techniques were not as effective; for example, Fathom has 304 false positives for sign-up pages, which results in a low precision score of 0.36. We found these false positives were Fathom misclassifying newsletter and contact forms as sign-up forms. Another source of false positives was Fathom classifying 209 pages as both login and sign-up pages. In contrast, our Autofill-based approach did not allow pages to be both login and sign-up pages; it could only be one or the other (or neither). Of the 23 manually analyzed pages that had both login and sign-up functionality, Autofill classified 22 as either login or sign-up (and classified one page as a non-authentication page), which affected its accuracy.

### 4.2 Fingerprinting by Page Type

**Rates of Fingerprinting.** Table 5 shows that 9.3% of the total pages our crawler visited were flagged as fingerprinting. This figure is slightly lower than the 10% rate from a 2021 study [33]. However, when we consider only login and sign-up pages, the percentage rises slightly to 10.2%, with the majority of scripts being attributed to third-party sources.

**Third Party Fingerprinting Scripts.** We find that websites often treat their login pages differently than their sign-up pages. Of the 1,902 domains that include third-party fingerprinting scripts on a login or sign-up page, 914 (48.05%) fingerprint on only the login page. The remaining sites are evenly split between fingerprinting only on the sign-up page (473 domains, 24.87%) and fingerprinting on both pages (515 domains, 27.08%). This pattern holds across popularity rankings; for every site rank bucket, 48-56% of the websites that fingerprint on an authentication page only do so on the login

page. While it is difficult to determine the intent behind fingerprinting, this suggests that sites are fingerprinting to enhance user security.

However, when sites fingerprint on both their login and sign-up pages, they almost exclusively use the same fingerprinting scripts on both. Of these 515 domains, 505 (98.06%) used scripts from the same set of third parties (based on the domains that provide the scripts). Of the 10 domains that had a different set of third parties, six include additional third parties on one of the pages, and one appears to use content served from the same entity that uses distinct domains (bmcdn5.com and bmcdn6.com). Only two domains appeared to use scripts from different entities. For example, sunglasshut.com includes a potentially fingerprinting script from smct.io on its sign-up page, but not on its login page. This script is from the company intent.ly (per the Disconnect entity list [4]), which advertises services for measuring and increasing customer conversions [6].

**Tracking vs Non-Tracking Scripts.** We labeled fingerprinting scripts as tracking or non-tracking using classifications from uBlock Origin Core [11], which rely on blocklists such as EasyList and EasyPrivacy. We found that home pages have the highest rate of tracking at 61.46%, compared to login pages at 50.50% and sign-up pages at 55.65%. The rate of tracking for authentication pages is surprising; websites are more likely to fingerprint on their login pages but less likely to use tracking scripts than sign-up pages. We checked how often each fingerprinting attribute (canvas, canvas fonts, WebRTC, and AudioContext) is called by tracking and non-tracking scripts. We found the prevalence of each attribute is approximately the same except for canvas font fingerprinting, which is used by non-tracking scripts more often (2.81% vs 0.73%).

**Fingerprinting APIs.** Lin et al. found that sites performed canvas fingerprinting an order of magnitude more than canvas fonts, WebRTC, or AudioContext fingerprinting [38]. We similarly investigate the rates of each type of fingerprinting for login, sign-up, and homepages in Figure 4. We confirm the finding from Lin et al. that canvas fingerprinting is the most popular type; in fact, nearly every homepage that performs fingerprinting engages in canvas

| All pages | | | | Login and sign-up pages | | | |
|---|---|---|---|---|---|---|---|
| Entity | Domain/Script | Category | Num. sites | Entity | Domain/Script | Category | Num. sites |
| Adscore Tech. | adsco.re | Ad Motivated Tracking Ad Fraud | 1,907 | Signifyd Inc. | signifyd.com | Fraud Prevention | 239 |
| - | wpadmngr.com | Advertising | 1,418 | Alibaba Group | aeis.alicdn.com/AWSC/WebUMID/1.93.0/um.js * | Marketing Analytics | 201 |
| Signifyd Inc. | signifyd.com | Fraud Prevention | 1,414 | Amazon Tech. | ssl-images-amazon.com | Marketing Advertising | 171 |
| Bounce Exchange | bounceexchange.com | Ad Motivated Tracking Advertising | 1,330 | Bounce Exchange | bounceexchange.com | Ad Motivated Tracking Advertising | 159 |
| InsurAds | insurads.com | Analytics | 1,229 | Sift Science, Inc. | sift.com | Fraud Prevention | 148 |
| Alibaba Group | aeis.alicdn.com/AWSC/WebUMID/1.93.0/um.js * | Marketing Analytics | 959 | FingerprintJS | cdnjs.cloudflare.com/ajax/libs/fingerprintjs2/2.1.2/fingerprint2.min.js | Fraud Prevention Analytics | 144 |
| Rambler Holding | top100.ru | Audience Measurement | 913 | Amazon Tech. | d38xvr37kwwhcm.cloudfront.net/js/grin-sdk.js | Marketing Advertising | 139 |
| Benhauer | salesmanago.pl | Customer Engagement | 112 | CHEQ AI Tech. | clickcease.com | Fraud Prevention | 118 |
| CHEQ AI Tech. | clickcease.com | Fraud Prevention | 719 | Rambler Holding | top100.ru | Audience Measurement | 113 |
| - | franecki.net | Marketing Analytics | 589 | Benhauer | salesmanago.pl | Customer Engagement | 112 |

**Table 4: The list of primary fingerprinting domains and related entities where at least one fingerprinting attempt was detected during a crawl conducted in August 2023. *Some entities may have multiple associated scripts.**

|  | Homepages | Inner pages | Login | Sign-up |
|---|---|---|---|---|
| **All** | 8,067 (8.5%) | 40,828 (9.2%) | 4,872 (9.2%) | 2,737 (12.5%) |
| **3rd party** | 4,639 (4.9%) | 24,701 (5.6%) | 2,294 (4.3%) | 1,539 (6.8%) |

**Table 5: The overall count of unique web pages where fingerprinting attempts were observed. Inner pages include login and sign-up pages.**

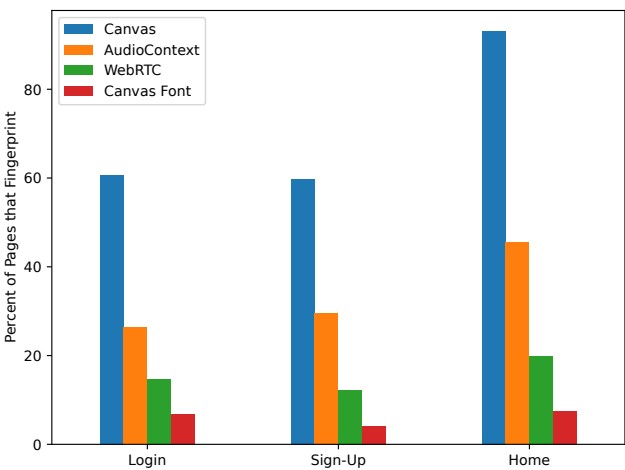

**Figure 4: For login, sign-up, and homepages that perform fingerprinting, we plot the percent of pages that call each fingerprinting API.**

fingerprinting (93.10%). Interestingly, we find similar frequencies of the types of fingerprinting performed by login and sign-up pages.

This supports our earlier finding that when sites fingerprint on both login and sign-up pages, they use the same fingerprinting scripts.

### 4.3 Comparison to Prior Results

Compared to a previous study by Durey et al. that examined the presence of fingerprinting scripts on checkout, basket, and authentication pages, we found fewer fingerprinting scripts on authentication pages [27]. In the previous study, they reported the detection of fingerprinting scripts on sign-up pages (31.1%), login pages (23.4%), and homepages (23.0%). Our results, on the other hand, show lower percentages: 9.2% for login pages, 8.5% for homepages, and 12.5% for sign-up pages. Nevertheless, the trend of observing more fingerprinting scripts on sign-up pages compared to login pages, and more scripts on login pages than homepages, remains consistent.

Several distinctions exist between our study and the prior one that explain the different fingerprinting rates. First, there is a disparity in the number of websites examined: we analyzed 100,000 sites while they only assessed login and sign-up pages from 446 domains. Furthermore, Durey et al. chose those 446 domains because they are more likely to collect sensitive personal information (e.g. sites for job searching or dating) or financial information (e.g. gambling, e-commerce); thus these sites are more likely to take care to prevent fraud. Another dissimilarity lies in the method employed for detecting fingerprinting scripts. We used a detection algorithm based on [33] that relies on just four primary browser attributes (Canvas, WebRTC, Canvas Font, and AudioContext fingerprinting).

We conducted an additional examination of the security organizations mentioned in the previous study by scrutinizing their domains in our dataset. Notably, we observed that one of the most prominent domains, sift.com, also ranked among the top fingerprinting domains in our list. While other security companies (Nudata Security - nudatasecurity.com, Simility - simility.com) also appeared in our results, they were observed on only a limited number of pages.

| | At Least 1 Page | Homepage | Login | Sign-up |
|---|---|---|---|---|
| **Top 1K** | 25.75% | 22.24% | 14.73% | 8.82% |
| **1K-5K** | 19.01% | 17.10% | 8.36% | 5.70% |
| **5K-10K** | 16.30% | 15.02% | 6.63% | 4.14% |
| **10K-50K** | 12.61% | 11.61% | 5.42% | 3.08% |
| **50K-100K** | 10.60% | 9.85% | 4.57% | 2.59% |

**Table 6: Percent of websites that perform fingerprinting on various page categories, grouped by popularity according to the CrUX dataset.**

Lin et al. measured how many of the Alexa top 20K websites perform fingerprinting on login pages [38]. They found 11.5K login pages for the Alexa top 20K (5,736 for the top 10K and 5,791 for rank 10K-20K), and found that the majority check basic device attributes such as the navigator and window objects. Using the criteria for fingerprinting from FP-Inspector [33] (i.e. the presence of canvas, canvas font, WebRTC, and AudioContext fingerprinting), Lin et al. found that 18% of login pages performing fingerprinting. [4]

There are a few potential explanations for why we find a lower rate of fingerprinting than Lin et al. [38]. First, we study a larger pool of websites (100K) than Lin et al (20K). Prior work has found that more popular websites are more likely to perform fingerprinting [33], so our fingerprinting rate will naturally decline as we consider more websites (i.e. include less popular websites). We present the rate of fingerprinting for each site-rank bucket in Table 6. As this table shows, we are able to confirm the finding that more popular websites are more likely to fingerprint. Interestingly, we find that this trend also holds for each type of page.

## 4.4 Potential Usage for Anti-Fraud

It is difficult to infer the intent behind the use of fingerprinting scripts. However, there are some clues that suggest websites may be using fingerprinting for anti-fraud purposes.

As indicated in Table 4, the most common fingerprinting domain on authentication pages, `signifyd.com`, is associated with a fraud prevention company. Additionally, we observed another script from a well-known fraud prevention company, served on `sift.com`, appearing among the top-10 fingerprinting domains on authentication pages. This suggests that browser fingerprinting is employed for purposes beyond tracking on authentication pages.

Through a small-scale manual analysis, we also found that disabling fingerprinting scripts broke login functionality. Due to the scalability challenges, we conducted a manual inspection of 30 websites where fingerprinting attempts were detected on login pages. We visited these websites using the JShelter browser extension [7], which can blocks fingerprinting scripts. Our findings revealed that the login functionality on `deezer.com` and `hepsiburada.com` (both of which have 12M+ active users) experienced disruptions when

fingerprinting scripts were blocked, as depicted in Appendix D. Specifically, on Deezer's login page, the login button became non-responsive, preventing users from signing in. Similarly, on Hepsiburada, attempting to log in from the homepage resulted in the login page failing to load.

It is also possible for websites to use fingerprinting for both anti-fraud and advertising (including ad-driven tracking) simultaneously. For instance, a widely used third-party script on 7% of authentication pages is from the aforementioned websites `sift.com` and `siftscience.com`; these are associated with a single fraud prevention company [17]. However, when we manually examined this script, we noticed that the users' fingerprints were sent to `hexagon-analytics.com`, which is controlled by the analytics company Hexagon Data [15].

## 5 LIMITATIONS

Similar to other research in web measurement, our study faces several limitations in terms of representativeness and coverage. For instance, websites may identify our crawler as an automated bot and treat it differently than genuine traffic. Although we depend on TRC's anti-bot measures [22], which provide some mitigation against bot detection, we recognize that their effectiveness may be limited [56].

The applicability and efficacy of our fingerprinting approach may also be limited. We rely on the technique developed by Englehardt et al. [33], which intentionally focuses on only four browser APIs.

Finally, our link detection method exclusively considers hyperlinks represented by the <a> element. However, certain login and sign-up forms may only become visible when triggered by user interaction with an HTML element. Our methodology may overlook these types of form. In addition, we do not attempt to fill out forms, and so we may miss additional multi-step authentication forms.

## 6 CONCLUSION

Browser fingerprinting, which is often associated with online tracking, is also sometimes used for user security by preventing account breaches, detecting bots, and thwarting cookie hijacking. Understanding the security implications of fingerprinting is crucial as mitigations are implemented. Our study fills a gap by examining the security aspects of fingerprinting at a large scale, particularly on login and sign-up pages. We introduced a highly accurate ML model (96-98% recall and precision) that successfully identified login (52,805) and sign-up (21,988) pages among 100,000 popular websites. Fingerprinting attempts were detected on 9.2% of these pages, slightly higher than the 8.9% rate across all pages. Notably, some of the top invasive fingerprinting scripts on login and sign-up pages were associated with fraud prevention companies, indicating diverse motivations for fingerprinting. We also show that when websites fingerprint on authentication pages, they are far more likely to only fingerprint on their login page. However, when they fingerprint on both login and sign-up pages, they almost always use the same set of fingerprinting scripts. Our contributions include an empirical web measurement study, a precise machine learning model, and practical tools (See Appendix A) for detecting login and sign-up pages. These findings illuminate the multifaceted role of browser fingerprinting beyond user tracking.

---

[4]Lin et al. find that the majority of the login pages they identified perform some form of fingerprinting such as accessing the navigator or window objects [38]. We use a more conservative definition of fingerprinting as explained in Section 3.2. Fortunately, Lin et al. describe the percentage of the login pages and homepages that perform each type of fingerprinting that we consider, and so we compare these rates. Lin et al. find that 2,133 websites perform canvas fingerprinting on their login pages, and they identified 11,527 login pages, yielding a fingerprinting rate of 18.50%.

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

## A  CODE AND DATA

The source code for both web crawlers (one dedicated to feature extraction, as discussed in §3, and another for the detection of login and sign-up URLs when given the homepage URL) and the browser add-on, along with the dataset derived from our research, will be publicly available.

**Browser Add-on** Through the integration of our ML-based classifier for login and sign-up pages within the extension, we have enabled the capability to determine whether a given page is a login or sign-up page. Additionally, the outcome of this classification is presented on the user interface of the extension.

**Login & Sign-up Links Detector** For forthcoming studies that necessitate the collection of login and sign-up URLs for the investigation of security and privacy concerns, we have developed a web crawler (an extended version of the TRC). This specialized crawler is devised to retrieve login and sign-up pages when provided with the homepage URL by using our ML model. The functioning of this crawler involves an initial assessment of whether the landing page is a login and sign-up page. If not, it proceeds to gather potential login and sign-up links using the method explained in § 3.5. Subsequently, it visits these links sequentially, halting when either a login and sign-up page is detected, or the list of links is exhausted. The output of the crawler comprises a list enumerating the located login and sign-up pages.

## B  FINGERPRINTING DETECTION HEURISTICS

Here are the heuristics employed in this study to identify fingerprinting attempts. Initially proposed by Englehardt and Narayanan [28], these heuristics were subsequently refined by Iqbal et al. [33].

**Canvas Fingerprinting:** A script is classified as a canvas fingerprinting script based on the following criteria:

(1) The script uses the canvas element to write text using methods such as `fillText` or `strokeText` and applies styling using methods like `fillStyle` or `strokeStyle` within the rendering context.
(2) The script invokes the `toDataURL` method to extract the canvas image.
(3) The script does not make use of the `save`, `restore`, or `addEventListener` methods on the canvas element.

**WebRTC Fingerprinting:** A script is identified as a WebRTC fingerprinting script according to the following conditions:

(1) The script invokes methods like `createDataChannel` or `createOffer` within the WebRTC peer connection.

(2) The script calls methods such as `onicecandidate` or `localDescription` within the WebRTC peer connection.

**Canvas Font Fingerprinting:** A script qualifies as a canvas font fingerprinting script based on the following criteria:

(1) The script sets the `font` property on a canvas element to utilize more than 20 different fonts.
(2) The script invokes the `measureText` method of the rendering context more than 20 times.

**AudioContext Fingerprinting:** A script is identified as an AudioContext fingerprinting script according to the following conditions:

(1) The script invokes any of the following methods within the audio context: `createOscillator`, `createDynamicsCompressor`, `destination`, `startRendering`, and `oncomplete`.

## C  LOGIN & SIGN-UP PAGE FEATURES

(1) **Form Features**
  (a) hasLogin: Checks all forms on the page for attributes containing a regular expression pattern contains several translations of terms associated with 'login'.
  (b) hasRegister: Checks all forms on the page for attributes containing a regular expression pattern contains several translations of terms associated with 'sign-up'.
  (c) hasNewsletter: Checks all forms on the page for attributes containing a regular expression pattern contains several translations of terms associated with 'newsletter'.
  (d) hasForgot: Checks all forms on the page for attributes containing a regular expression pattern contains several translations of terms associated with 'forgot'.

(2) **Button Features**
  (a) hasLoginInAttributes: Checks all buttons on the page for attributes containing a regular expression pattern contains several translations of terms associated with 'login'.
  (b) hasLoginInAttributesOnAForm: Checks all buttons on a form for attributes containing a regular expression pattern contains several translations of terms associated with 'login'.
  (c) hasLoginInTextContent: Checks all buttons on a page for its text content containing a regular expression pattern contains several translations of terms associated with 'login'.
  (d) hasLoginInTextContentOnAForm: Checks all buttons on a form for its text content containing a regular expression pattern contains several translations of terms associated with 'login'.
  (e) hasRegisterInAttributes: Checks all buttons on the page for attributes containing a regular expression pattern contains several translations of terms associated with 'register'.
  (f) hasRegisterInAttributesOnAForm: Checks all buttons on a form for attributes containing a regular expression pattern contains several translations of terms associated with 'register'.
  (g) hasRegisterInTextContent: Checks all buttons on a page for its text content containing a regular expression pattern contains several translations of terms associated with 'register'.

(h) hasRegisterInTextContentOnAForm: Checks all buttons on a form for its text content containing a regular expression pattern contains several translations of terms associated with 'register'.

(i) hasNewsletterInAttributes: Checks all buttons on the page for attributes containing a regular expression pattern contains several translations of terms associated with 'newsletter'.

(j) hasNewsletterInAttributesOnAForm: Checks all buttons on a form for attributes containing a regular expression pattern contains several translations of terms associated with 'newsletter'.

(k) hasNewsletterInTextContent: Checks all buttons on a page for its text content containing a regular expression pattern contains several translations of terms associated with 'newsletter'.

(l) hasNewsletterInTextContentOnAForm: Checks all buttons on a form for its text content containing a regular expression pattern contains several translations of terms associated with 'newsletter'.

(m) hasNextInAttributes: Checks all buttons on the page for attributes containing a regular expression pattern contains several translations of terms associated with 'next'.

(n) hasNextInAttributesOnAForm: Checks all buttons on a form for attributes containing a regular expression pattern contains several translations of terms associated with 'next'.

(o) hasNextInTextContent: Checks all buttons on a page for its text content containing a regular expression pattern contains several translations of terms associated with 'next'.

(p) hasNextInTextContentOnAForm: Checks all buttons on a form for its text content containing a regular expression pattern contains several translations of terms associated with 'next'.

(q) hasForgotInAttributes: Checks all buttons on the page for attributes containing a regular expression pattern contains several translations of terms associated with 'forgot'.

(r) hasForgotInAttributesOnAForm: Checks all buttons on a form for attributes containing a regular expression pattern contains several translations of terms associated with 'forgot'.

(s) hasForgotInTextContent: Checks all buttons on a page for its text content containing a regular expression pattern contains several translations of terms associated with 'forgot'.

(t) hasForgotInTextContentOnAForm: Checks all buttons on a form for its text content containing a regular expression pattern contains several translations of terms associated with 'forgot'.

(u) hasNextButtonCloseToUsername: Evaluates the text content of all buttons on a page using a regular expression pattern that includes various translations of terms related to 'next' aiming to identify the presence of a nearby username field.

(v) hasNextButtonCloseToUsernameOnAForm: Evaluates the text content of all buttons on a form using a regular expression pattern that includes various translations of terms related to 'next' aiming to identify the presence of a nearby username field.

(w) hasLoginButtonCloseToUsername: Evaluates the text content of all buttons on a page using a regular expression pattern that includes various translations of terms related to 'login' aiming to identify the presence of a nearby username field.

(x) hasLoginButtonCloseToUsernameOnAForm: Evaluates the text content of all buttons on a form using a regular expression pattern that includes various translations of terms related to 'login' aiming to identify the presence of a nearby username field.

(y) hasSignupButtonCloseToUsername: Evaluates the text content of all buttons on a page using a regular expression pattern that includes various translations of terms related to 'sign-up' aiming to identify the presence of a nearby username field.

(z) hasSignupButtonCloseToUsernameOnAForm: Evaluates the text content of all buttons on a page using a regular expression pattern that includes various translations of terms related to 'sign-up' aiming to identify the presence of a nearby username field.

(3) **Anchor Features**

(a) hasForgotInAttributes: Checks all anchor on the page for attributes containing a regular expression pattern contains several translations of terms associated with 'forgot'.

(b) hasForgotInAttributesOnAForm: Checks all anchor on a form for attributes containing a regular expression pattern contains several translations of terms associated with 'forgot'.

(c) hasForgotInTextContent: Checks all anchor on a page for its text content containing a regular expression pattern contains several translations of terms associated with 'forgot'.

(d) hasForgotInTextContentOnAForm: Checks all anchor on a form for its text content containing a regular expression pattern contains several translations of terms associated with 'forgot'.

(4) **Label Features**

(a) hasRememberMeInAttributes: Checks all anchor on the page for attributes containing a regular expression pattern contains several translations of terms associated with 'remember me'.

(b) hasRememberMeInAttributesOnAForm: Checks all anchor on a form for attributes containing a regular expression pattern contains several translations of terms associated with 'remember me'.

(c) hasRememberMeInTextContent: Checks all anchor on a page for its text content containing a regular expression pattern contains several translations of terms associated with 'remember me'.

(d) hasRememberMeInTextContentOnAForm: Checks all anchor on a form for its text content containing a regular expression pattern contains several translations of terms associated with 'remember me'.

(e) hasNewsletterMeInAttributes: Checks all anchor on the page for attributes containing a regular expression pattern contains several translations of terms associated with 'newsletter'.

(f) hasNewsletterMeInAttributesOnAForm: Checks all anchor on a form for attributes containing a regular expression pattern contains several translations of terms associated with 'newsletter'.

(g) hasNewsletterMeInTextContent: Checks all anchor on a page for its text content containing a regular expression pattern contains several translations of terms associated with 'newsletter'.

(h) hasNewsletterMeInTextContentOnAForm: Checks all anchor on a form for its text content containing a regular expression pattern contains several translations of terms associated with 'newsletter'.

(5) **Header Features**

(a) hasLoginInAttributes: Checks all headers on the page for attributes containing a regular expression pattern contains several translations of terms associated with 'login'.

(b) hasLoginInAttributesOnAForm: Checks all headers on a form for attributes containing a regular expression pattern contains several translations of terms associated with 'login'.

(c) hasLoginInTextContent: Checks all headers on a page for its text content containing a regular expression pattern contains several translations of terms associated with 'login'.

(d) hasLoginInTextContentOnAForm: Checks all headers on a form for its text content containing a regular expression pattern contains several translations of terms associated with 'login'.

(e) hasRegisterInAttributes: Checks all headers on the page for attributes containing a regular expression pattern contains several translations of terms associated with 'register'.

(f) hasRegisterInAttributesOnAForm: Checks all headers on a form for attributes containing a regular expression pattern contains several translations of terms associated with 'register'.

(g) hasRegisterInTextContent: Checks all headers on a page for its text content containing a regular expression pattern contains several translations of terms associated with 'register'.

(h) hasRegisterInTextContentOnAForm: Checks all headers on a form for its text content containing a regular expression pattern contains several translations of terms associated with 'register'.

(i) hasNewsletterInAttributes: Checks all headers on the page for attributes containing a regular expression pattern contains several translations of terms associated with 'newsletter'.

(j) hasNewsletterInAttributesOnAForm: Checks all headers on a form for attributes containing a regular expression pattern contains several translations of terms associated with 'newsletter'.

(k) hasNewsletterInTextContent: Checks all headers on a page for its text content containing a regular expression pattern contains several translations of terms associated with 'newsletter'.

(l) hasNewsletterInTextContentOnAForm: Checks all headers on a form for its text content containing a regular expression pattern contains several translations of terms associated with 'newsletter'.

(m) hasForgotInAttributes: Checks all headers on the page for attributes containing a regular expression pattern contains several translations of terms associated with 'forgot'.

(n) hasForgotInAttributesOnAForm: Checks all headers on a form for attributes containing a regular expression pattern contains several translations of terms associated with 'forgot'.

(o) hasForgotInTextContent: Checks all headers on a page for its text content containing a regular expression pattern contains several translations of terms associated with 'forgot'.

(p) hasForgotInTextContentOnAForm: Checks all headers on a form for its text content containing a regular expression pattern contains several translations of terms associated with 'forgot'.

(6) **Checkbox Features**

(a) hasNewsletterInAttributes: Checks all checkbox on the page for attributes containing a regular expression pattern contains several translations of terms associated with 'newsletter'.

(b) hasNewsletterInAttributesOnAForm: Checks all checkbox on a form for attributes containing a regular expression pattern contains several translations of terms associated with 'newsletter'.

(c) hasRememberMeInAttributes: Checks all checkbox on the page for attributes containing a regular expression pattern contains several translations of terms associated with 'remember me'.

(d) hasRememberMeInAttributesOnAForm: Checks all checkbox on a form for attributes containing a regular expression pattern contains several translations of terms associated with 'remember me'.

(7) **Input Features**

(a) hasLoginInAttributes: Checks all input on the page for attributes containing a regular expression pattern contains several translations of terms associated with 'login'.

(b) hasLoginInAttributesOnAForm: Checks all input on a form for attributes containing a regular expression pattern contains several translations of terms associated with 'login'.

(c) hasRegisterInAttributes: Checks all input on the page for attributes containing a regular expression pattern contains several translations of terms associated with 'register'.

(d) hasRegisterInAttributesOnAForm: Checks all input on a form for attributes containing a regular expression pattern contains several translations of terms associated with 'register'.

(e) hasNewsletterInAttributes: Checks all input on the page for attributes containing a regular expression pattern contains several translations of terms associated with 'newsletter'.

(f) hasNewsletterInAttributesOnAForm: Checks all input on a form for attributes containing a regular expression pattern contains several translations of terms associated with 'newsletter'.

(g) hasAnyEmail: Check whether there is any email field.

(h) hasAnyUsername: Check whether there is any username field.

(i) hasAnyPEmailOnAForm: Check whether there is any email field on a form.

(j) hasAnyUsernameOnAForm: Check whether there is any username field on a form.

(k) hasMultipleInputs: Check whether there is multiple input fields.

(8) **Password Features**

(a) hasLabelOrAriaLabelOrPlaceholderContainsConfirm: Checks all password's label, aria label and placeholders on the page for attributes containing a regular expression pattern contains several translations of terms associated with 'confirm'.

(b) hasLabelOrAriaLabelOrPlaceholderContainsConfirmOnAForm: Checks all password's label, aria label and placeholders on a form for attributes containing a regular expression pattern contains several translations of terms associated with 'confirm'.

(c) hasLabelOrAriaLabelOrPlaceholderContainsCurrent: Checks all password's label, aria label and placeholders on the page for attributes containing a regular expression pattern contains several translations of terms associated with 'current'.

(d) hasLabelOrAriaLabelOrPlaceholderContainsCurrentOnAForm: Checks all password's label, aria label and placeholders on a form for attributes containing a regular expression pattern contains several translations of terms associated with 'current'.

(e) hasLabelOrAriaLabelOrPlaceholderContainsNew: Checks all password's label, aria label and placeholders on the page for attributes containing a regular expression pattern contains several translations of terms associated with 'new'.

(f) hasLabelOrAriaLabelOrPlaceholderContainsNewOnAForm: Checks all password's label, aria label and placeholders on a form for attributes containing a regular expression pattern contains several translations of terms associated with 'new'.

(g) hasAnyPasswordField: Check whether there is any password field.

(h) hasMultiplePasswordFields: Check whether there is multiple password fields.

(9) **Div Features**

(a) hasAlreadyHaveAnAccount: Checks all divs on the page for attributes containing a regular expression pattern contains several translations of terms associated with 'already have an account'.

(b) hasAlreadyHaveAnAccountOnAForm: Checks all divs on a form for attributes containing a regular expression pattern contains several translations of terms associated with 'already have an account'.

(c) hasDontHaveAnAccount: Checks all divs on the page for attributes containing a regular expression pattern contains several translations of terms associated with 'don't have an account'.

(d) hasDontHaveAnAccountOnAForm: Checks all divs on a form for attributes containing a regular expression pattern contains several translations of terms associated with 'don't have an account'.

(e) hasNewsletter: Checks all divs on the page for its tect content containing a regular expression pattern contains several translations of terms associated with 'newsletter'.

(10) **URL Features**

(a) hasResetInURL: Checks the URL of the page for attributes containing a regular expression pattern contains several translations of terms associated with 'reset password'.

(b) hasNewsletterInURL: Checks the URL of the page for attributes containing a regular expression pattern contains several translations of terms associated with 'newsletter'.

# D WEB PAGE BEHAVIORS WHEN FINGERPRINTING SCRIPTS ARE BLOCKED

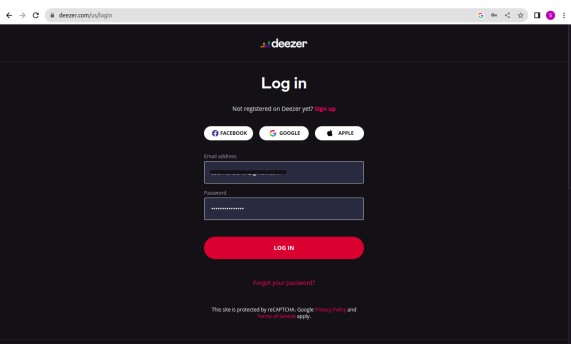

(a) Deezer's login form when JShelter was disabled.

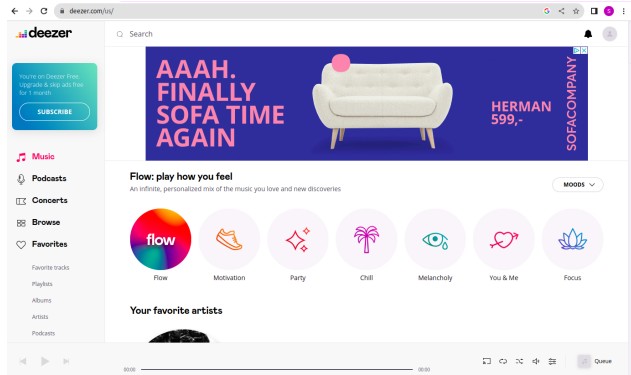

(b) Deezer's homepage when a user is signed in and fingerprinting scripts are not blocked

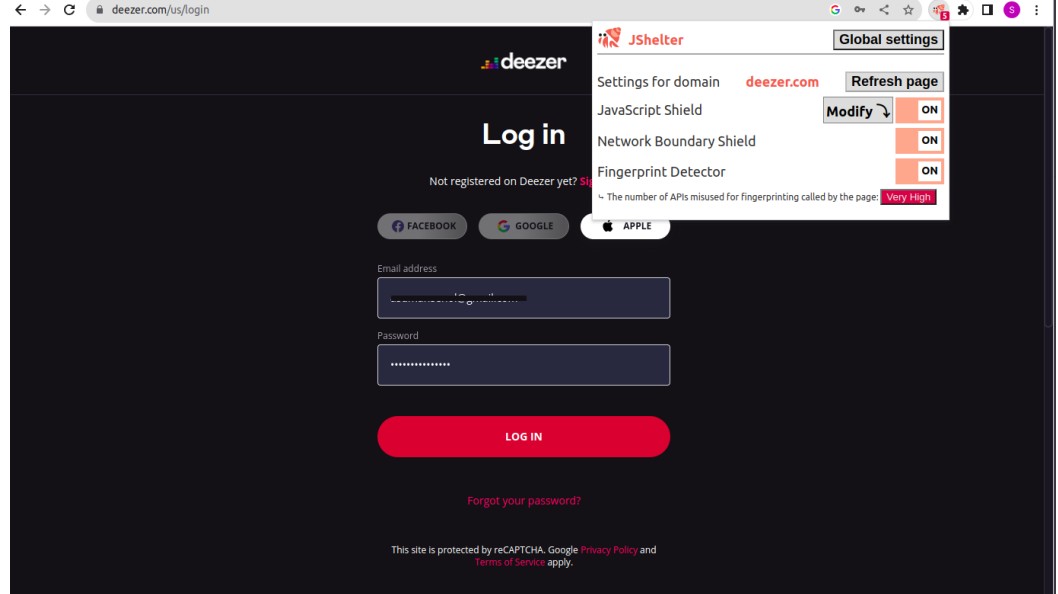

(c) When JShelter is activated and blocks fingerprinting scripts, the functionality of Deezer's login form is affected, causing the login buttons to behave abnormally.

**Figure 5: Effect of fingerprinting script blocking on Deezer's login page. Deezer's login form behaves abnormally when JShelter is activated and blocks fingerprinting scripts ©, causing disruptions in the functionality of login buttons.**

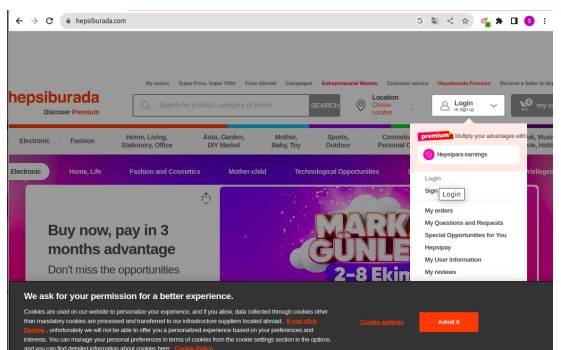

(a) The login form of Hepsiburada with JShelter deactivated.

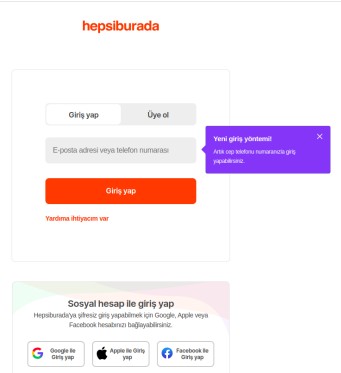

(b) Upon clicking the login button in Figure (a), users are directed to the Hepsiburada login page with JShelter disabled.

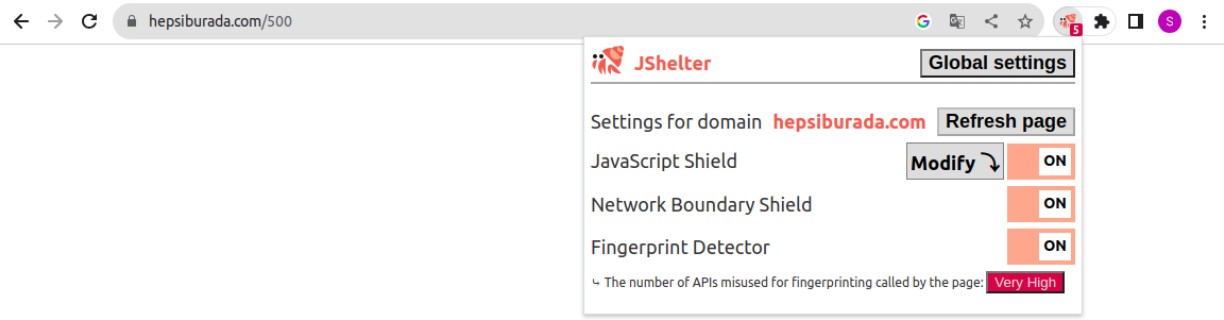

(c) The Hepsiburada login page fails to load when the extension is active, and the extension has disabled fingerprinting scripts.

Figure 6: Hepsiburada's Login Page Behavior with JShelter Activation

