# OpenReview forum: "The Double Edged Sword: Identifying Authentication Pages and their Fingerprinting Behavior"
_ACM.org/TheWebConf/2024/Conference — TheWebConf24_

### Official Review · Reviewer_GGe2 · 2023-11-19

**Novelty:** 5
**Technical Quality:** 5

**Review:**

This paper introduced a machine learning model to classify sign-up, sign-in, and non-authentication web pages, and then performed a browser fingerprinting detection on sign-up and sign-in pages to learn its uses. The features for web page classification were extracted from the HTML DOM, using the "new" headless mode of Puppeteer crawler. The paper also made a comparison with existing approaches for the web page classification. In the second part of the paper, a fingerprinting detection was performed on top 100K domains, leveraging existing work [28] and its follower [33]. The paper provided discussions on fingerprinting rate, third party fingerprinting scripts, tracking scripts, and fingerprinting APIs as well as a comparison with prior work.

Pros:
1. A promising classifier for sign-up and sign-in pages.
2. A large scale measurement of browser fingerprinting on sign-up and sign-in pages in top 100K domains.

Cons are mainly about the second half of the paper - browser fingerprinting detection.
1. This part merely used techniques in exising work [28, 33]. It is unclear about its technical contribution.
2. There are three work published recently, as [1, 2, 3] listed below, about the fingerprinting detection. They all highlighted that the flow of browser attributes plays a significant role in browser fingerprinting detection, no matter using static analysis or dynamic analysis/tainting. However, this work focused on the invocation of the fingerprinting APIs only while ignored the flow. In addition, researchers have found that objects/APIs under JavaScript objects such as navigator, screen, and window are heavily used for fingerprinting [3], which was not considered in this work. This work leveraged only the APIs for canvas, audio, etc., based fingerprinting. It is not clear about how this work address these two concerns.
3. I am very confused about the first paragraph in Sectoin 3.5 - it just used regular expressions on a link URL to check if that link points to a sign-up or sign-in page. The big question is, what Section 3.1 is used for? Shouldn't it be used here? Please correct me if I misunderstood something.
4. At the end of Sectoin 4.1, it said "our ... approach did not allow pages to be both login and sign-up pages". However, later in the same paragraph, there indeed were 22 pages for both login and sign-up. This problem needs to be addressed.
5. According to the findings in the first paragraph of Section 4.2, about 10.2% of login and sign-up pages are performing browser fingerprinting, consistent to the findings of recently work such as [33] studing all types of pages. It would be better to highlight the importance of studying browser fingerprinting on only the login and sign-up pages.
6. It is unclear about the differences in the findings between this work and [38].

Minor mistakes.
    1. In the second paragraph of Section 2.3, is it Figure 6 or Figure 1?

[1] Junhua Su, Alexandros Kapravelos: Automatic Discovery of Emerging Browser Fingerprinting Techniques. WWW 2023.
[2] Alexander Sjösten, Daniel Hedin, Andrei Sabelfeld: EssentialFP: Exposing the Essence of Browser Fingerprinting. EuroS&P Workshops 2021.
[3] Rui Zhao, FProbe: The Flow-Centric Detection and a Large-Scale Measurement of Browser Fingerprinting, ICCCN 2023.

**Questions:**

Cons raised up above.

**Reviewer Confidence:**

4: The reviewer is certain that the evaluation is correct and very familiar with the relevant literature

**Scope:**

4: The work is relevant to the Web and to the track, and is of broad interest to the community

---

### Official Review · Reviewer_5X5H · 2023-11-23

**Novelty:** 6
**Technical Quality:** 6

**Review:**

The paper addresses the problem of identitying fingerprinting in auth pages. To this end, the authors, propose a new machine learning-based that automatically detects authentication pages, outperforming other methods. Then, they measure they evaluate the prevalence of fingerprinting scripts across login pages compared to "normal" sites, showing that they differ in the scripts they include,

The paper addresses a relevant and yet omitted in the literature about fingerprinting. The results obtained (specially the measurement) highlights the necessity of additional strategies for fingerprinting.

**Questions:**

Is any of the fingerprinting used "legitimately" as an auth factor misused? That could be interesting to explore (a subsample).

**Reviewer Confidence:**

4: The reviewer is certain that the evaluation is correct and very familiar with the relevant literature

**Scope:**

4: The work is relevant to the Web and to the track, and is of broad interest to the community

---

### Official Review · Reviewer_dNAR · 2023-11-23

**Novelty:** 4
**Technical Quality:** 6

**Review:**

This paper provides a measurement study of browser fingerprinting on login and sign-up web pages. The authors state that existing fingerprinting defenses do not take into account legitimate use of fingerprinting, such as during user authentication, which can provide an additional layer of security. To understand the usage of fingerprinting on authentication pages, they first developed a new model to identify login and sign-up pages with 96-98% precision and recall and applied it on a large set of 100K web pages. The findings include a slightly higher prevalence of fingerprinting scripts on authentication pages (9.2% vs. 8.9%), as well as the difference in fingerprinting behavior, such as third-parties used for fingerprinting.

The content of the paper is visibly divided into two parts: (1) A novel ML-powered methodology to detect login and sign-up pages, (2) Measurements study of fingerprinting scripts on such authentication pages vs. main pages of websites. While I see a clear novelty in the first part and indeed the proposed model looks performing clearly better than the prior works, the measurement results in (2) look rather expected and overall confirming previous studies, and I feel a lack of concrete discussion on how existing fingerprinting defenses can be improved using these take-aways. For example, the confirmed finding that there are more fingerprinting scripts on sign-up pages than on login pages sounds expected as websites are likely doing a user conversion on those pages. Overall, I have the following questions to clarify the methodology and take-aways:
- For this measurement study, how important is it to use the improved login / sign-up identification method?  In other words, any trends or crucial examples you were not able to discover with less login / sign-up page coverage?
- One of the stated findings is that fingerprinting scripts on authentication pages are more often tagged as “fraud prevention”, which implies that existing fingerprinting detection methods tend to separate those. Do you have an evaluation of how often authentication-related fingerprinting scripts are miscategorized and blocked? Or whether web users can allow “fraud prevention” scripts via exceptions? Do you see any opportunity to distinguish such fingerprinting scripts vs. trackers?

In addition, I have the following technical questions:
- The bottleneck of identifying login and sign-up pages becomes the fact whether a right page is crawled and analyzed. The authors mention that they first navigate to the homepage of each website and then visit up-to 15 inner pages linked on the homepage and matched with regex patterns. Could you describe the details on what patterns are used and how did you decide on the number of links?
- Did you take into account fingerprinting on SSO pages?
- Could you clarify the stated “+83% F1-score” in the abstract? Meaning, how it was calculated as I do not see it being mentioned in the main text.

At the same time, I liked the small-scale experiments reported in 4.4. I would encourage the authors to expand those and describe take-ways in depth. The authors also state that they will release to the public a browser extension to identify login and sign-up pages - it would be nice to describe more details on that browser extension and how it can be used.

**Questions:**

Selected questions:
- For this measurement study, how important is it to use the improved login / sign-up identification method? In other words, any trends or crucial examples you were not able to discover with less login / sign-up page coverage?
- One of the stated findings is that fingerprinting scripts on authentication pages are more often tagged as “fraud prevention”, which implies that existing fingerprinting detection methods tend to separate those. Do you have an evaluation of how often authentication-related fingerprinting scripts are miscategorized and blocked? Or whether web users can allow “fraud prevention” scripts via exceptions? Do you see any opportunity to distinguish such fingerprinting scripts vs. trackers?
- The bottleneck of identifying login and sign-up pages becomes the fact whether a right page is crawled and analyzed. The authors mention that they first navigate to the homepage of each website and then visit up-to 15 inner pages linked on the homepage and matched with regex patterns. Could you describe the details on what patterns are used and how did you decide on the number of links?

**Reviewer Confidence:**

4: The reviewer is certain that the evaluation is correct and very familiar with the relevant literature

**Scope:**

4: The work is relevant to the Web and to the track, and is of broad interest to the community

---

### Official Review · Reviewer_Y2bJ · 2023-11-24

**Novelty:** 5
**Technical Quality:** 5

**Review:**

Rebuttal response
----------------------------
Thank you for answering my questions in detail - both those explicitly phrased and those left more implicit. I have not updated my review nor scoring, since my position is unchanged: I am positive about this paper.

Summary
----------------------------
This paper investigates use of fingerprinting as an authentication-strengthening measure.


Strong points
----------------------------
- Investigating and measuring overlooked benign use of browser fingerprinting
- On-topic, in scope, and of interest


Weak points
----------------------------
- Weak distinction between auth-strengthening fingerprinters and non-auth-strengthening fingerprinters
  This is your motivation, but the measurement doesn't quite live up to this.


Overall evaluation
----------------------------
I think this is a paper that will be of interest to the WWW-SEC community. It is decently executed and written.


Comments for authors
----------------------------
- Abstract, "To the best of our knowledge"

  Well, FPBlock does achieve this, but it's not quite intentional. That is, it ensures that each first party gets its own
  unique fingerprint. That is its goal, which happens to overlap with the sought behaviour here. This is more of a side
  effect and is (to the best of my memory) not mentioned at all in the FPBlock paper -- so no need to add a reference;
   just wanted you to be aware.

- pg 6, "While it is difficult":

  If sites would want to enhance user security, they would track users
  throughout their site in order to establish a much better profile. Restricting
  themselves to login pages is like tying shoes with one hand: results aren't
  that great.

- pg 6, "Tracking vs non-tracking scripts":

  It isn't sufficiently clear whether authentication would benefit (in your opinion) from tracking or not.

- pg 8, "anti-fraud"

  One popular category of fighting online fraud revolves around fighting online adclick fraud. This is not
  the same as preventing authentication fraud. (The fingerprinters for the former tend to be much
  more evolved than for the latter, which paints a bleak picture of where priorities lie.)

**Questions:**

- Could you elaborate (beyond what's currently in the paper) how to tell whether a fingerprinting script is used to strengthen authentication?

**Ethics Review Description:**

-

**Reviewer Confidence:**

4: The reviewer is certain that the evaluation is correct and very familiar with the relevant literature

**Scope:**

4: The work is relevant to the Web and to the track, and is of broad interest to the community

---

### Decision · Program_Chairs · 2024-01-22

**Decision:**

Accept

**Comment:**

The reviewers agreed that this research makes some useful contributions to the literature, in particular in its ability to discriminate sign-up pages and login pages. The technical execution is competent, pushing the paper above the acceptance bar. The main criticisms lie in the large adoption of existing and relatively standard techniques, with limited novel contributions that others might build upon. The paper can be improved through a more in-depth discussion of related work, that the authors promised to do in their rebuttal.

 ---